# Breast Cancer Stem Cells: Signaling Pathways, Cellular Interactions, and Therapeutic Implications

**DOI:** 10.3390/cancers14133287

**Published:** 2022-07-05

**Authors:** Lei Wang, Zeng Jin, Rohan P. Master, Chandra K. Maharjan, Madison E. Carelock, Tiffany B. A. Reccoppa, Myung-Chul Kim, Ryan Kolb, Weizhou Zhang

**Affiliations:** 1Department of Pathology, Immunology and Laboratory Medicine, College of Medicine, University of Florida, Gainesville, FL 32610, USA; lei.wang@ufl.edu (L.W.); zengjin@ufl.edu (Z.J.); rohan.master@ufl.edu (R.P.M.); cmaharjan@ufl.edu (C.K.M.); madison.carelock@ufl.edu (M.E.C.); treccoppa@ufl.edu (T.B.A.R.); my.kim@ufl.edu (M.-C.K.); ryankolb@ufl.edu (R.K.); 2Immunology Concentration, Biomedical Graduate Program, College of Medicine, University of Florida, Gainesville, FL 32610, USA; 3Cancer Biology Concentration, Biomedical Graduate Program, College of Medicine, University of Florida, Gainesville, FL 32610, USA; 4Department of Biology, College of Liberal Arts & Sciences, University of Florida, Gainesville, FL 32610, USA; 5UF Health Cancer Center, University of Florida, Gainesville, FL 32610, USA

**Keywords:** breast cancer stem cells, signaling pathways, cell interactions, clinical trials

## Abstract

**Simple Summary:**

Cancer stem cells (CSCs) in breast cancer have been identified for almost two decades. Many outstanding discoveries have established the important functions of CSCs in breast cancer progression, metastasis, and resistance to therapy. As the defining feature of CSCs, stemness is induced and maintained by several important signaling pathways. Targeting these pathways is inevitably challenging because they are also critically involved in normal stem cells. Here, we will summarize the literature on breast cancer stem cells, including major signaling pathways, cellular interactions within the tumor microenvironment (TME), and potential therapeutic implications.

**Abstract:**

Breast cancer stem cells (BCSCs) constitute a small population of cells within breast cancer and are characterized by their ability to self-renew, differentiate, and recapitulate the heterogeneity of the tumor. Clinically, BCSCs have been correlated with cancer progression, metastasis, relapse, and drug resistance. The tumorigenic roles of BCSCs have been extensively reviewed and will not be the major focus of the current review. Here, we aim to highlight how the crucial intrinsic signaling pathways regulate the fate of BCSCs, including the Wnt, Notch, Hedgehog, and NF-κB signaling pathways, as well as how different cell populations crosstalk with BCSCs within the TME, including adipocytes, endothelial cells, fibroblasts, and immune cells. Based on the molecular and cellular activities of BCSCs, we will also summarize the targeting strategies for BCSCs and related clinical trials. This review will highlight that BCSC development in breast cancer is impacted by both BCSC endogenous signaling and external factors in the TME, which provides an insight into how to establish a comprehensively therapeutic strategy to target BCSCs for breast cancer treatments.

## 1. Introduction

Breast cancer is the most common malignancy in women worldwide and the majority of breast cancer patients (~70–80%) in the early stages are curable [1]. Currently, the challenges for breast cancer patients are metastatic diseases and relapse after treatments, both of which are directly correlated with an increased mortality rate in patients at late stages (20–30% of patients at the early stage will die of relapse with metastasis) [2]. Apart from breast cancer stage, another important factor for the prognosis of breast cancer is the cancer subtype, because breast cancer is a heterogeneous disease at both the pathological and molecular levels [3]. The classic pathological classification of breast cancer is based on the expression of hormone receptors and human epidermal growth factor receptor 2 (HER2): luminal breast cancer (estrogen receptor and progesterone receptor, ER/PR-positive), HER2-positive (HER2-positive, ER/PR-negative), and triple negative (ER-, PR-, HER2-negative) breast cancer (TNBC). The classification is largely reflective of their gene signatures, which further divide breast cancer into intrinsic molecular subtypes by PAM50 signature: luminal A, luminal B, HER2-enriched, basal-like breast cancer (BLBC), and normal-like [4]. TNBC and BLBC are largely overlapping with around 70% of BLBC being TNBC. The claudin-low subtype introduces another level of complexity for subtyping breast cancer [5], which may have a different cell-of-origin and reflect a poor prognosis [6]. To achieve better clinical outcomes, therapeutic strategies based on stages and subtypes are considered the first step toward precision therapy. The treatments for breast cancer patients in the early stage are mainly surgery, radiotherapy, and chemotherapy. For luminal breast cancer, patients are treated with hormonal agents or HER-2 antibodies to prevent further metastasis [7,8]. In metastatic breast cancer, targeted drug therapies, such as CDK4, mTOR, and receptor tyrosine kinase (RTK) inhibitors, are used to treat some patients [9,10,11]. Additionally, immunotherapy is a relatively novel and promising strategy to enhance the host immune system to eliminate cancer cells. Thus far, pembrolizumab is the only FDA-approved immune checkpoint inhibitor (ICI) for metastatic TNBC, which blocks programmed cell death protein 1 (PD-1) to enhance the T-cell-mediated anti-tumor responses [12]. Even though these treatments can be effective in controlling breast cancer, some cancer cells are able to evade treatments and survive, leading to cancer relapse. Accumulating evidence indicates that BCSCs are relatively resistant to current therapeutics and may play a pivotal role in breast cancer progression and relapse after treatment [13]. Therefore, eliminating BCSCs is an emerging effort for therapeutic development.

BCSCs are a small population of breast cancer cells with the abilities of self-renewal and tumorigenesis [14]. The origin of BCSCs is controversial, echoing the ‘the chicken or the egg’ causality dilemma in breast cancer. Currently, there are two different hypotheses. One hypothesis is that BCSCs originate from more differentiated non-stem cells that are undergoing epithelial-to-mesenchymal transition (EMT) [15]. The environmental stress on differentiated non-stem cells induces genetic and epigenetic alterations resulting in the emergence of BCSCs [16,17,18,19]. The other hypothesis is that BCSCs are the cancerous mammary stem cells and progenitor cells that harbor the oncogenic somatic mutations during non-malignant stem cell differentiation [20]. Notably, although tumor initiates from different environments, one common feature of these two hypotheses is that BCSCs have variable mutations. Distinct tumor environments shape the molecular regulations of BCSCs and drive interactions between BCSCs and other cell populations [21].

BCSCs are identified from heterogeneous breast cancer cells by various cell surface markers. Al-Hajj, et al. first identified and isolated CD44^+^CD24^−/low^-lineage^−^ cancer cells in breast cancer patients, and they found that this population of breast cancer cells exhibited tumorigenic capacity [13]. Later on, a study showed that stem-like breast cancer cells have a high expression level of aldehyde dehydrogenase (ALDH) which is correlated with a poor prognosis [22]. Subsequent studies have proved that the CD44^+^CD24^−/low^ and ALDH1^+^ population has significant BCSC activity, including self-renewal, metastasis, and drug resistance [23,24,25,26]. Apart from CD44^+^CD24^−/low^ and ALDH1^+^, various markers have been reported to identify and isolate BCSCs in human and mouse breast cancer, which supports further research on BCSCs and provides a premise for clinical BCSC targeting (Table 1).

Here, we highlight the molecular signaling pathways that control the function of BCSCs, such as self-renewal, and their involvement in tumorigenesis and metastasis. We will also review the interactions between BCSCs and the other cell populations within the TME. Furthermore, we will summarize relevant clinical trials and preclinical research that are devoted to eliminating BCSCs.

## 2. Intrinsic Molecular Activities of BCSCs

### 2.1. Signaling Pathways

#### 2.1.1. Wnt Signaling Pathway

The Wnt signaling pathway regulates the self-renewal capacity and the differentiation potential of normal mammary stem cells [39]. The Wnt signaling pathway is commonly dysregulated in various cancers mostly due to loss of function mutation/deletion in the tumor suppressor gene *Adenomatous polyposis coli* (*APC*). It has been shown that constitutive activation of Wnt/β-catenin signaling transforms a population of luminal progenitor cells and endows them with tumorigenic capacity [40]. Presumably, BCSCs can retain Wnt/β-catenin activation to continuously fuel breast cancer progression [40,41,42,43]. Non-canonical Wnt5A and Wnt5B have also been reported to have a significant function in maintaining the stemness of BCSCs [44,45]. There have been studies supporting the idea that blocking canonical Wnt/β-catenin signaling significantly reduces the number of BCSCs in breast cancer [42,46,47].

Several proteins either involved in the Wnt signaling pathway or interacting with proteins in the Wnt signaling pathway can mediate BCSC self-renewal and metastasis. ΔNp63 is a transcription factor that enhances normal mammary stem cell (MaSC) activities by upregulating the expression of Frizzled Class Receptor 7 (Fzd7)—a Wnt receptor. BCSCs utilize the same mechanism to promote tumorigenesis in TNBC [48]. Limb-Bud-and-Heart (LBH) is a co-transcription factor of Wnt/β-catenin target genes that participates in activating the stem cell transcription program in breast cancer cells, further leading to cancer metastasis. Previous research has shown that inhibition of Cadherin 11 (CDH11)—the expression of which is correlated with shorter survival in TNBC—can downregulate β-catenin, which suppresses the canonical Wnt signaling pathway, thereby inhibiting the cancer stem-like phenotype in TNBC [49,50,51]. Several recent studies have proved XB130 overexpression in malignancies. Functionally, XB130 promotes EMT—a prerequisite cellular process for BCSC induction [52]—and accelerates tumor initiation via the Wnt/β-catenin signaling pathway [53].

The Wnt signaling pathway can also be regulated by dysregulation of micro-RNAs (miR) in BCSCs. Lethal-7 (let-7) miRs are significantly reduced in the MCF-7 breast cancer cell line and restoring them can inhibit breast cancer cell proliferation [54]. One recent study has shown that let-7b and let-7c were inversely correlated with ERα expression that stimulates BCSC self-renewal via the Wnt signaling pathway. The overexpression of let-7c in MCF-7 cells significantly decreases tumor growth [55]. miR-600 can target stearoyl-Coenzyme A desaturase 1 (SCD1), an enzyme required to activate Wnt signaling, and previous research has found that silencing miR-600 results in BCSC expansion whereas overexpressing it can reduce BCSC self-renewal [56]. miR-140 plays an important tumor-suppressive role by regulating the Wnt signaling pathway. miR-140-5p is downregulated in BCSCs and restoring miR-140-5p prevents the proliferation of BCSCs. Mechanistically, miR-140-5p can suppress Wnt1 to prevent BCSC proliferation [57,58]. miR-125b is overexpressed in Snail-induced BCSCs and promotes BCSC propagation and chemoresistance by upregulating the Wnt signaling pathway [59]. miR-142 effectively binds to *APC* mRNA and restores the RNA-induced silencing complex, leading to the activation of the canonical Wnt signaling pathway. Knockdown of miR-142 significantly inhibits the organoid formation of BCSCs and delays tumor initiation [41]. Overexpression of miR-31 inhibits several Wnt antagonists, such as Dickkopf-1(Dkk1), which helps BCSCs expand at the expense of differentiation in vivo [60] (Figure 1a).

#### 2.1.2. Notch Signaling Pathway

The activation of Notch signaling pathways can be observed in normal and malignant stem cells. By the crosstalk with other signaling pathways, such as the JAK-STAT pathway, normal stem cells can maintain the homeostasis between self-renewal, differentiation, and proliferation [61]. Similarly, the aberrant activation of Notch signaling pathways in BCSCs supports their self-renewal, expansion, and metastasis [62]. Therefore, the Notch signaling pathway is considered a critical signaling pathway that influences BCSC fate [63].

The Notch signaling pathway can be activated when Notch transmembrane receptors bind with their ligands, such as Delta-like canonical Notch ligand (Dll) 1/3/4 and Jagged (JAG) 1/2 that are secreted by neighboring cells or themselves. Treatment of breast cancer cell lines with IgG-69, a Dll1 antibody, can effectively inhibit mammosphere formation [64]. Interestingly, the Notch signaling pathway seems to be quite specific to BCSCs for BLBC. Physiological Notch activation leads to the differentiation of luminal progenitors, a process that is tightly regulated by the breast cancer type 1 susceptibility protein (BRCA1) via JAG1 expression [65]. The activation of Notch pathways seems to be suppressive for spheroid formation within the luminal BCSCs, as knockdown of JAG1 leads to increased tumorospheres [65]. In contrast, in BCSCs from BLBC, JAG1 can be included by a NF-κB-dependent mechanism, leading to juxtacrine activation of Notch signaling and expansion of BCSCs [66,67]. JAG2 can be strongly induced under a hypoxic environment, which promotes EMT and increases the number of BCSCs [68].

In addition to the direct ligand-receptor activation, other mechanisms have been studied. One study demonstrated that bone morphogenetic protein 4 (BMP-4) activates the Notch signaling pathway via a Smad4-dependent manner in MCF-10 cells, thereby promoting the EMT and advancing the cancer stem cell properties [69]. MAP17, a small protein overexpressed in cancer cells, can activate the Notch signaling pathway by hijacking a Notch antagonist, NUMB, and subsequently elevates the stem gene expression and tumorosphere number in breast cancer [70]. Similarly, miR-146a blocks the function of NUMB to help generate BCSCs [71]. Alternatively, breast cancer stemness can be induced by the interaction between the Notch signaling pathway and other signaling pathways. For example, the Notch–CCR7 signaling axis promotes stemness in MMTV-PyMT breast cancer cells [72] (Figure 1b).

#### 2.1.3. Hedgehog (HH) Signaling Pathway

The HH signaling pathway facilitates embryonic development and cell differentiation. The dysregulation of the HH signaling pathway is implicated in TNBC and HER2-positive breast cancer [73] and is persistently activated in both normal mammary stem cells and BCSCs [74,75,76].

There are three HH homologs in HH signaling pathways, including Desert (DHH), Indian (IHH), and Sonic (SHH). The Sonic HH signaling pathway is the most well-studied in breast cancer. In the transgenic MMTV-ErbB2 (HER2-positive) mouse model, ΔNp63 supports breast cancer stemness by inducing the elevation of the SHH signaling cascade, inducing the expression of Smoothened (SMO), Protein Patched Homolog 1 (PTCH1), and GLI Family Zinc Finger 2(GLI2) [77]. The constitutive activation of the HH signaling pathway results in breast tumorigenesis and invasiveness [78]. GLI Family Zinc Finger 1 (GLI1)-mediated signaling and VEGF receptor neuropilin (NRP2) signaling form an autocrine loop to continuously promote tumor initiation in TNBC. NRP2 overexpression stimulates the upregulation of α6β1 integrins and GLI1. GLI1 can induce polycomb complex protein BMI-1, a stem cell regulator, which further increases NRP2 and α6β1 integrins activity, thereby accelerating tumor initiation [79]. SMO is a key receptor in the SHH signaling pathway and regulates GLI proteins to promote BCSC differentiation. By contrast, BCSC activity can also be regulated by SMO-independent HH signaling. Forkhead box C1 (FOXC1), an EMT transcription factor specifically expressed in BLBCs, binds to and activates GLI2, which regulates BCSC enrichment via the SMO-independent HH signaling pathway [80] (Figure 1c).

#### 2.1.4. NF-κB Signaling Pathway 

The NF-κB signaling pathway plays a crucial role in cancer cell survival, inflammation, and immunity [81,82]. In breast cancer, the activation of the NF-κB signaling pathway can be used as a prognostic marker and it is involved in cancer cell proliferation, differentiation, and invasiveness [83,84]. In addition, multiple studies have supported that the NF-κB signaling pathway promotes the stemness of BCSCs and accelerates both tumorigenesis and invasiveness by promoting the EMT process [85,86].

BCSC function and fate are regulated by both canonical and non-canonical NF-κB signaling pathways. The canonical NF-κB signaling pathway can be stimulated by diverse tumor necrosis factor (TNF) superfamily members. Using TNFα as the prototypical example for NF-κB activation, TNFα engagement with TNF receptors (TNFRs) induces inhibitor-κ kinases (IKK) α/β/γ ternary complex activation that can induce subsequent phosphorylation, polyubiquitination, and degradation of IκBα. Following degradation of IκBα, the RelA/p50 dimer, otherwise sequestered by IκBα to the cytosol, is released and translocated into the nucleus for transcriptional regulation of target genes. Dll1^+^ quiescent BCSCs with activated canonical NF-κB signaling pathways drive resistance to chemotherapy in breast cancer treatment. Inhibition of either Dll1 or NF-κB signaling can enhance the sensitivity of breast luminal tumors to chemotherapy [87]. One study has shown that the IKK complex-driven canonical NF-κB signaling pathway accelerates self-renewal in BLBC. Inhibition of IKKβ or knockdown of different IKK and NF-κB subunits results in a decreased tumorosphere formation of SUM149 [88]. Biglycan (BGN) regulates the activation of NF-κB signaling pathways. Knockdown of BGN in BCSCs has been shown to result in the reduction of tumorigenic phenotypes, lower metastatic potential, and attenuation of NF-κB signaling pathways [89].

The non-canonical NF-κB signaling pathway is also mainly initiated by TNF superfamily members. Using the receptor activator of nuclear factor kappa-B ligand (RANKL) as the prototypical example of a non-canonical NF-κB signaling pathway, RANKL—produced by either luminal mammary epithelial cells [90] or tumor-infiltrating regulatory T cells [91]—can lead to persistent activation of NF-κB-inducing kinase (NIK) via protein stabilization [91,92]. NIK-mediated IKKα activation [92] induces several important signaling events that are known to be important for BCSC self-renewal, expansion, and mammary tumorigenesis via the non-canonical NF-κB or other signaling pathways [91,92,93]. TNF-α upregulates transcriptional co-activator with PDZ-binding motif (TAZ) via the non-canonical NF-κB signaling pathway [94]. TAZ is a transcription factor that promotes BCSC self-renewal capacity in human breast cancer cell lines [95]. High levels of NIK expression increase breast cancer cell tumorigenicity and upregulate BCSC markers, such as aldehyde dehydrogenase 1 family, member A1 (ALDH1A1) [86]. Interestingly, both canonical and non-canonical NF-κB signaling pathways have been implicated to activate the juxtacrine Notch signaling pathway via elevating JAG1 expression, leading to an expansion of BCSC populations in breast cancer [66,67] (Figure 1d).

#### 2.1.5. Other Signaling Pathways

Recent research has discovered that other signaling pathways also play non-negligible roles in BCSC development. Epidermal growth factor receptor (EGFR)-mediated PI3K/AKT, MAPK, and STAT3 signaling pathways are hyperactivated in the CD44^+^/CD24^−^ BCSC population in MDA-MB-231, a human triple negative breast cancer cell line [96,97]. Inhibition of EGFR significantly induces the mesenchymal–epithelial transition and increases the cancer cell sensitivity to chemotherapies. Among the HER family members, HER3 is typically considered to mediate PI3K/AKT signaling and drive tumorigenesis [98]. Additionally, HER2 drives BCSC self-renewal and tumorigenesis through the activation of PI3K/AKT signaling with HER3-dependent or -independent mechanisms [99,100,101]. JAK/STAT3 signaling regulates BCSC self-renewal and chemoresistance by activating fatty acid β-oxidation [102]. Transforming growth factor β (TGFβ) signaling has been reported in multiple breast cancer subtypes, including claudin-low breast cancer, TNBC, and HER2-positive breast cancer [103,104,105]. TGFβ signaling also increases the metastatic capacity of CD44^+^CD24^−^ BCSC rather than CD44^+^CD24^−^ non-stem-like cells. Treating breast cancer cell lines with TGFβ increases the expression levels of BCSC markers, including Nanog, Pou5f1, and Sox2 [106].

### 2.2. Other Transcription Factors

In addition to the transcriptional factors involved in the aforementioned signaling pathways, some key transcription factors that can dictate stem cell fate include Oct-4, Nanog, and Sox-2. These transcription factors are consistently activated in CSCs to maintain their self-renewal capacity [107,108,109]. Comparing the Oct-4^high^ cell population with the Oct-4^low^ cell population in the 4T1 mouse breast cancer model, researchers found that Oct-4^high^ 4T1 cells prefer to form tumorospheres and have higher levels of stem cell markers [110]. Further evidence showed that isolated Oct-4^high^ cells from murine MC4-L2 cells have BCSC features [111]. Nanog is a known pluripotent transcription factor in embryonic stem cells [112]. In an inducible Nanog transgenic mouse model, Nanog promotes breast cancer tumorigenesis and metastasis [113]. High-level Nanog expression upregulates EMT-related genes in BCSCs [109]. Sox-2 is another transcription factor that maintains stemness. Sox-2 activation is directly associated with the spheroid formation in BCSCs [114,115]. One study has demonstrated that the FBXW2–MSX2–Sox-2 axis regulates the BCSC properties and drug resistance [116]. Interestingly, tamoxifen resistance might be driven by Sox-2 in BCSCs via downstream activation of Wnt signaling pathways [117]. The scheme of intrinsic signaling pathways related to BCSCs is summarized in Figure 1.

## 3. Cellular Crosstalk between BCSCs and Different Cell Populations in the TME

### 3.1. Adipocytes

Adipocytes are the major cell type in the breast with roughly 90% of the organ consisting of adipose tissue. In breast cancer, studies have highlighted the role of adipocytes in promoting cancer progression by creating an inflammatory environment through the release of cytokines, chemokines, and growth factors, enhancing the cancer cells to acquire treatment resistance [118,119]. Previous research has shown that adipocytes play a role in assisting breast cancer initiation, and they further accelerate breast cancer progression due to their role under a hypoxic environment [120]. With a low oxygen level, adipocytes begin to express Interleukin-6 (IL-6) to support BCSC survival by activating the Notch signaling pathway [121]. Furthermore, adipocyte-derived IL-6 promotes the spheroid formation of BCSCs by stimulating NF-κB and STAT3 signaling pathways in HER2-positive breast cancer [122,123]. Adipocyte-derived chemokine (C-C motif) ligand 2 (CCL-2) plays dual functions in promoting breast cancer development by regulating breast cancer stemness and creating an immunosuppressive environment via the recruitment of macrophages through the CCL-2-IL-1β signaling axis [124,125,126,127].

Adipokines, such as leptin and adipsin, are correlated with breast cancer progression, and as such, their expression levels are elevated in obesity-associated breast cancer [128,129]. Leptin promotes breast cancer development by stimulating breast epithelial cell stemness and drives BCSC enrichment to promote tumorigenesis in an obesity-driven TNBC model [130,131]. Additionally, leptin has been shown to induce several transcription factors that promote the stemness of BCSCs including Nanog, Oct-4, and Sox-2 [132,133,134]. Furthermore, leptin can interact with the leptin receptor (LEPR) on BCSCs to activate the JAK/STAT3 signaling pathway. *Carnitine palmitoyltransferase 1B* (*CPT1B*), a STAT3 target gene, is upregulated in BCSCs and encodes a critical enzyme for fatty acid β-oxidation, which leads to self-renewal of BCSCs and chemoresistance [102]. Adipsin, also known as complement factor D, can increase the BCSC population in a breast patient-derived xenograft (PDX) model [135]. It functions as a protease and converts complement C3 into small fragments C3a and C3b. With the blockade of C3aR, the mammosphere formation induced by adipsin is abolished. Adipsin can also stimulate the adipose tissue-derived stem cells to produce hepatocyte growth factor (HGF) and promote the mammosphere formation of BCSCs [136]. Adipocytes play a critical role in promoting BCSC formation through the secretion of multiple cytokines and adipokines.

### 3.2. Fibroblasts

Cancer-associated fibroblasts (CAFs) are a dynamic and abundant cell type in the TME with diverse functions. They are involved in extracellular matrix remodeling, crosstalk with the immune cells, and interactions with cancer cells. CAFs also promote BCSC proliferation through either cell-to-cell contact or secreted factors. Co-culturing MCF-7 breast cancer cells with CAFs induces higher mammosphere formation and stem cell-related gene expression including Wnt1, Notch1, β-catenin, chemokine receptor (Cxcr) type 4, Sox-2, and aldehyde dehydrogenase 3 family member A1 (Aldh3a1), compared to those with the MCF-7 breast cancer cells co-cultured with normal mammary fibroblasts [137,138]. Additionally, CAFs can be activated by HH ligand stimulation, and they provide a niche for supporting BCSC function and growth in a mouse TNBC model [139]. When Hs-578Bst cells, the normal mammary fibroblasts, are co-cultured with BCSCs, the fibroblasts secreted an elevated amount of extracellular matrix metalloproteinase inducer (EMMPRIN), also known as CD147 [140]. EMMPRIN can increase the number of CD44^+^CD24^−^ BCSCs, and the upregulation of EMMPRIN expression is positively correlated with STAT3 and Hypoxia-inducible factors-1α (HIF-1α) levels. Knockdown of both STAT3 and HIF-1α reduces the BCSC ability to form mammospheres. Certain subtypes of CAFs, including CD10^+^GPR77^+^ CAFs, can enhance the formation of ALDH1^+^ BCSCs through secreted IL-6 and IL-8 [141]. Additionally, CAFs with IL-7 expression can promote the clonogenic potential of E0771 cells, suggesting that CAF-derived IL-7 is important for maintaining a niche for BCSCs [142]. Furthermore, miR-221 derived from CAF-secreted microvesicles is utilized by breast cancer cells to ultimately transform non-cancer stem cells into CD133^high^ BCSCs. The production of miR-221 within CAFs is mediated by the IL6-STAT3 pathways [143]. Lastly, CAFs secrete a high level of CCL2, which promotes BCSC formation via Notch signaling [144]. The high plasticity of CAFs allow them to differentiate into multiple subtypes, which secret cytokines to enhance the generation and maintenance of BCSCs.

### 3.3. Endothelial Cells and Vasculogenic Mimicry

Endothelial cells are traditionally known for their role in blood vessel formation, but recent studies highlighted their involvement in enhancing the formation of BCSCs. Co-culturing endothelial cells with breast cancer cells can significantly increase the number of mammospheres [145]. Endothelial JAG1 activates the Notch1 receptor on breast cancer cells to induce Zinc finger E-box-binding homeobox 1 (ZEB1) expression, which subsequently leads to the expression of stem cell markers [146]. The loss of endothelial JAG1 attenuates their ability to induce BCSCs. Additionally, endothelial cells expressing mesenchymal markers can enhance the tumor survival, stemness, and invasiveness of the breast cancer cells [147]. When BCSCs interact with arteriolar endothelial cells via the lysophosphatidic acid (LPA)/protein kinase D (PKD-1) signaling pathway, it strengthens both endothelial cell differentiation and BCSC self-renewal by regulating CD36 transcription [148].

BCSCs can differentiate into endothelial-like cells or form blood vessels through vasculogenic mimicry to support tumor growth [149,150,151]. When human BCSCs are implanted in severe combined immunodeficient (SCID) mice, intratumoral human CD31-positive cells can be detected, indicating that the BCSCs may be able to differentiate into endothelial lineages in vivo [149]. BCSCs can differentiate into endothelial cells with the addition of vascular endothelial growth factors (VEGF) in vitro [149,152]. ZEB1 activation has been shown to promote the trans-differentiation of BCSCs into endothelial-like cells [153]. The knockdown of *Zeb1* by siRNA or miRNA can inhibit the tube network formation by BCSCs. Additionally, endothelial differentiation requires functional autophagy and the knockdown of autophagy-related 5 (Atg-5) protein severely inhibits this process [154]. VEGF-treated BCSCs can differentiate into endothelial-like cells through activation of ZBTB10, and the differentiated BCSCs can secrete a higher amount of VEGF, resulting in enhanced blood vessel formation. VEGF can bind to VEGF receptor 2 (VEGFR2) to activate the STAT3 pathway as well as to increase Myc and Sox-2 expression to enhance the stemness of the breast cancer cells [155]. NRP2 is a co-receptor of VEGF, and with VEGF activation, it can increase the number of BCSCs through TAZ-mediated inhibition of Rac GAP β2-chimaerin [156]. The VEGF-A/NRP-1 axis can contribute to BCSC stemness, and it is highly dependent on the Wnt/β-catenin pathway [157]. Both HIF-1α and VEGFR2 are important for the hypoxia-induced differentiation of BCSCs into endothelial cells. Apatinib, a tyrosine kinase inhibitor with high selectivity against VEGFR2, can significantly inhibit the vasculogenic mimicry formation by the BCSCs [158]. Both CD133^+^ and ALDH1^+^ BCSCs have an enhanced vasculogenic mimicry function compared to traditional BCSCs [150,151,159,160]. Endothelial cells promote BCSC production through surface protein interactions with the breast cancer cells. Even in the absence of endothelial cells, BCSCs can upregulate multiple pathways to initiate vasculogenic mimicry and supply the growth of the tumor. The crosstalk between stromal cells and BCSCs is illustrated in Figure 2.

### 3.4. Immune Cells

BCSCs use different mechanisms to drive their immune evasion. Similar to normal stem cells, BCSCs can either maintain a quiescent status to avoid elimination by effector immune cells or form an immunosuppressive TME by recruiting immunosuppressive populations, such as regulatory T (Treg) cells, tumor-associated monocytes/macrophages (TAMs), and tumor-infiltrated monocytic myeloid-derived suppressor cells (MDSCs) (Figure 3) [161,162].

BCSCs develop multiple strategies to escape immune surveillance [163]. Natural killer (NK) cells are a powerful group of innate immune cells with the anti-tumor surveillance function. One mechanism by which NK cells recognize and kill cancer cells is through the priming of NK cells via NK cell receptor-natural killer group 2, member D (NKG2D) [164]. The expression of MHC class I chain-related protein A and B (MICA and MICB), two ligands of NKG2D, is reduced in BCSCs because of miR20a dysfunction. Thus, BCSCs can escape the immune surveillance from NK cells, a process that is critical for lung metastasis [165]. BCSCs can slow the cell cycle by producing Dickkopf WNT signaling pathway inhibitor 1 (DKK1), which reduces the expression of UL16-binding protein 1 (ULBP) ligands for the NKG2D receptor, helping BCSCs increase their resistance to NK cell killing [166]. Non-lytic CD8 T cells fuel the self-renewal and tumorigenesis of BCSCs. One study has reported that the cognate interaction between non-lytic CD8 T cells and BCSCs increases the stem cell-like population in the MCF-7 breast cancer cell line [167]. Furthermore, BCSCs can decrease their sensitivity to anti-tumor cytokines. BCSCs suppress the expression of ligand-dependent nuclear receptor corepressor (LCOR), which can prime BCSCs to interferon (IFN) responses [168]. Moreover, effector T cells are inefficient in eradicating BCSCs due to the higher expression of programmed death-ligand 1 (PD-L1) on BCSCs as compared to non-stem-like breast cancer cells [169,170]. Myc expression positively correlates with the expression of BCSC markers, and it upregulates the expression of PD-L1 and CD47 which can both attenuate T-cell activity and circumvent the phagocytosis of macrophages [171,172]. In addition to inactivating T cells, PD-L1 and β-catenin signaling form a positive loop that mutually upregulates the expression levels of each other, resulting in the maintenance and expansion of BCSCs [173].

Another way that BCSCs escape immune surveillance is by decreasing antigen-presenting capacity to maintain their quiescence [174]. A genome-wide CRISPR/Cas9 screen discovered that the major histocompatibility complex (MHC-I) antigen processing pathway is silenced in BCSCs, which blocks tumor antigen presentation and T-cell activation [175]. Ligand-dependent co-repressor (LCOR) is known to be involved in normal and malignant breast stem cell differentiation. LCOR^low^ BCSCs are resistant to ICIs because of their reduced antigen presentation machinery [176].

Moreover, the immunosuppressive TME can be formed by the interaction between BCSCs and specific immune cell populations. Treg cells increase the number of ALDH1^+^ BCSCs and promote cancer cell spheroid formation. Treg cells regulate Sox-2 overexpression in BCSCs, which activates NF-κB-CCL-1 signaling that subsequently recruits more Tregs to the TME [177]. Recently, a study has profiled the molecular portraits from normal stem cells to CSCs by single-cell analysis during malignant transformation. This research has found that immune cell infiltrations may contribute to immunosuppressive features during tumorigenesis. One evidence to support their discovery is that the expression levels of cytokines in BCSCs are positively correlated with their receptors in immune cells. For example, high levels of chemokine (C-X-C motif) ligand (Cxcl) 1 and Cxcl16 in BCSCs correlate with high expression of their corresponding receptors, Cxcr2 in macrophages and Cxcr6 in T cells [178]. The upregulation of CD90 and EPH Receptor A4 (EphA4) in BCSCs attracts TAMs, which creates a BCSC niche through juxtracrine signaling [36]. In the primary tumor site, MDSCs induce the EMT phenotype of BCSCs to promote cancer cell dissemination. Subsequently, pulmonary granulocytic MDSCs reverse the EMT phenotype to the MET phenotype for metastatic cancer cell proliferation in the lung [179]. MDSCs also endow stemness to breast cancer cells via IL-6/STAT3 and NO/Notch crosstalk signaling [180]. Moreover, Lin 28 is highly expressed in BCSCs and promotes tumor stemness, metastasis, and invasion [181]. A recent study has shown that Lin 28 not only promotes the transformation of primary tumor cells to ALDH^+^ BCSCs but also recruits neutrophils and converts them to the N2 subtype, an immunosuppressive neutrophil population, to further enhance the immunosuppressive TME for BCSCs [182].

Apart from quiescence and an immunosuppressive status, BCSCs can also convert themselves to the metastatic cancer stem cells (MCSCs) by interacting with neutrophils. Leukotriene receptor (LTR)-positive MMTV-PyMT cancer cells have BCSC properties, such as high spheroid formation and tumorigenesis. LTR^+^ BCSCs respond to neutrophil-derived leukotriene and transform into MCSCs. Experiments have shown that MMTV-PyMT tumor cells cultured with the conditioned medium of lung infiltrating neutrophils have more metastatic initiation potential [183].

## 4. Developing Therapeutics for Eliminating BCSCs in Clinical Trials

Classic chemotherapy or targeted therapies aim to kill bulk cancer cells but are not ideal for eliminating BCSCs due to their slow propagation and/or resistance to those therapeutics. Eliminating BCSCs is a promising therapeutic strategy to prevent cancer relapse and overcome treatment resistance. However, specifically targeting BCSCs is still a challenge in breast cancer treatment since BCSCs share many of the same features and markers as normal mammary stem and progenitor cells. In the wake of developments in breast cancer treatment, some therapeutic strategies have exhibited certain effects on BCSC elimination. Multiple preclinical studies have shown initial promising results to eliminate BCSCs and some of them are in clinical trials [184,185].

### 4.1. Targeting Surface Markers

Specific surface proteins on BCSCs provide potential targets to eradicate BCSCs. In preclinical research, P245, a CD44 monoclonal antibody, effectively prevents TNBC development and relapse by inhibiting CD44^+^ BCSCs [186]. CD44-binding peptide, CD44BP, inhibits the formation of mammary stem cell spheres in a PEGDA gel culture system [187]. FK506-binding protein like (FKBPL) and its peptide derivative, AD-01, also exhibit anti-BCSC activity via the modulation of CD44 signaling [188]. Bivatuzumab mertansine, a CD44 monoclonal antibody–drug conjugate, has been tested in a phase I clinical trial (NCT02254005) but the result is unknown. CD133 is another potential marker for targeting BCSCs [189]. Using CD133-targeted nanoparticles (nanoparticles conjugated with anti-CD133 antibody) loaded with paclitaxel represses local tumor recurrence in a mouse model of breast cancer [190]. A clinical trial has been testing the expression of CD133 in mammary invasive ductal carcinoma and exploring the correlation between CD133 and current known clinicopathological parameters (NCT04873154). ALDH1 is a common BCSC marker, and researchers have made efforts to target ALDH1^+^ BCSCs in the laboratory setting. Quercetin inhibits BCSCs by downregulating ALDH1 [191]. Curcumin analogs target ALDH1 and GSK-3β to overcome chemoresistance in breast cancer [192]. Similar to CD133, clinical trials of ALDH-targeted therapies are also ongoing (NCT00949013, NCT01424865, NCT04581967). HER2 drives breast cancer cell stemness in both luminal and HER2-positive breast cancer [193,194,195,196], raising the potential to use anti-HER2 therapy in all breast cancers where BCSCs express HER2. As such, one clinical trial is to examine the use of BCSC markers as an indicator to evaluate the effectiveness of Trastuzumab, a HER2-specific monoclonal antibody that has been approved by the FDA for patients with HER2-positive cancers (Table 2).

### 4.2. Targeting the Crucial Signaling Pathway of BCSCs

The aberrant activation of signaling pathways in BCSCs is discussed in Section 2 (Intrinsic molecular activities of BCSCs). Targeting these aberrant signaling pathways has shown some anti-tumor effects in breast cancer. Some studies and clinical trials have shown that targeting aberrant signaling pathways also decreases the proportion of BCSCs and their self-renewal capacity [197,198,199] (Table 3).

#### 4.2.1. Targeting Wnt Signaling Pathway

Vantictumab is an antibody that can bind to frizzled receptors and suppresses the canonical Wnt signaling pathway in cancer cells. Vantictumab is in clinical trial 1b and the drug is used to treat patients with HER2-negative cancers. The result has shown an outcome of 33% partial response (NCT01973309) [200]. Foxy-5, a Wnt-5a agonist shown to inhibit breast cancer metastasis in a mouse TNBC model [201], is undergoing a phase I clinical trial for metastatic breast cancers (NCT02020291) [202]. Cirmtuzumab is a monoclonal antibody that targets ROR1, a receptor involved in non-canonical signaling. Cirmtuzumab has been shown to dedifferentiate cancer stem cells in leukemia [203]. Cirmtuzumab is also undergoing a phase I clinical trial for breast cancer treatment (NCT02776917).

#### 4.2.2. Targeting Notch Signaling Pathway

γ-secretase inhibitors (GSIs) are traditional Notch signaling inhibitors and have antineoplastic activity in clinical applications. GSIs decrease the number of BCSCs in MC1 and BCM-2147 PDX models by inhibiting the Notch signaling pathway [204]. In a clinical trial, a GSI PF-03084014 has been used in TNBC and metastatic breast cancer treatment. However, the outcome has shown only a small population with a complete response (CR) or partial response (PR) (13–20%). Thus, a single inhibitor treatment might not be an ideal option (NCT02299635). In another clinical trial, advanced breast cancer patients with a GSI MK-0752 treatment plus docetaxel have shown decreased CD44^+^CD24^−^ and ALDH^+^ BCSC populations and lower mammosphere formation from their biopsies [204,205] (NCT00106145, NCT00645333). In TNBC, Vitamin D compounds have been shown to eliminate BCSCs by inhibiting Notch1, Notch2, Notch3, JAG1, and JAG2 [199]. In a clinical trial, Vitamin D has been combined with melatonin to reduce the spread of cancer cells (NCT01965522).

#### 4.2.3. Targeting Hedgehog (HH) Signaling Pathway

The HH signaling pathway plays a key role in regulating the stem cell program and its dysregulation has been observed in CSCs to promote CSC self-renewal [206]. Taladegib is an antagonist of the HH-ligand cell surface receptor smothered (SMO) and has shown potential antineoplastic activity. Taladegib has been tested in a clinical trial against advanced and metastatic breast cancers (NCT02784795). However, Taladegib combined with Crenigacestat, an oral Notch and gamma-secretase inhibitor, has been terminated in a clinical trial because of its toxicity and the poor response from patients [207]. Itraconazole is an FDA-approved antifungal drug and has recently been repurposed as an SMO inhibitor. Itraconazole has been reported to increase the survival of TNBC patients (NCT00798135, NCT04712396) [208]. One disadvantage of SMO inhibitors is the resistance observed in the late stage of treatment [209]. GLIs are transcription factors downstream of the HH signaling pathway. A preclinical trial has shown that targeting GLI1 exhibits an anti-tumor effect and efficiently overcomes the tumor resistance caused by SMO inhibitors in breast cancers [210]. In terms of targeting HH signaling, although the current clinical activities are towards broad therapeutic effects in TNBC or HER-2 positive breast cancer—two BCSC-enriched breast cancer types—it is also promising to effectively eliminate BCSCs in metastatic and relapsed breast cancer.

### 4.3. Targeting the Components in the BCSC Microenvironment—VEGF

VEGF signaling regulates cancer progression by enhancing angiogenesis and vascular permeability [156], but has also been shown to drive BCSC self-renewal via VEGFR2/STAT3-mediated upregulation of Myc and Sox-2 [155,211]. Bevacizumab is a monoclonal antibody that blocks angiogenesis by inhibiting VEGF-A [212]. Bevacizumab is used in both research and clinical trials, but the treatment outcome is not consistent (NCT00016549, NCT01190345). In another clinical trial, bevacizumab has been combined with conventional chemotherapy to inhibit BCSC activity (NCT00095706). However, there is a study that has highlighted the potential of how bevacizumab could induce a hypoxic environment and further increase BCSC numbers (Table 4) [213].

### 4.4. Immunotherapy

#### 4.4.1. Vaccines

HIF-1α promotes cancer growth in low oxygen environments and the mechanisms through which HIF-1α helps BCSCs are described in low oxygen environments through mechanisms described in the previous section [141,158]. One recent study has shown that mice immunized by a HIF-1α peptide pool, namely the HIF-1α vaccine, have a higher level of HIF-1-specific IgG in sera, which neutralizes HIF-1α, enhances Th1 and Th2 immunity, and reduces tumor growth in TNBC and activities of BCSCs [214]. Digoxin, which can inhibit HIF-1α and block tumor growth [215], has shown therapeutic potential in a clinical trial (NCT01763931) for breast cancer treatment. Brachyury is overexpressed in TNBC and its expression is regulated by IL-8, a cytokine important to maintain BCSCs [216]. The use of Brachyury as a target for vaccine design was proposed to treat TNBC, especially by preventing the growth of stem-like cancer cells [217]. The Brachyury vaccine educates the host immune cells to target the Brachyury protein, and there have been clinical trials to test this vaccine against multiple subtypes of breast cancer (NCT02179515, NCT04296942, NCT04134312, NCT03384316). Some other anti-tumor vaccines in clinical trials are also optimized to enhance the immune response to specifically eliminate BCSCs. One clinical trial evaluates the multiantigen DNA plasmid-based vaccine, which is a cocktail vaccine including the proteins of BCSCs (NCT02157051). Another clinical trial examines the safety of one CSC vaccine, which contains high levels of IgG binding to CSCs and subsequently increases the immune response to CSC antigens (NCT02063893).

#### 4.4.2. CAR-T Cell Therapy

CAR-T cells are engineered T cells with the expression of specific antigen receptors which endows T cells with the capability to eliminate BCSCs by targeting their specific proteins [218]. CD133^+^ breast cancer cells have stem-like properties and CD133 has also been identified as a BCSC marker [189]. CD133 mRNA was transfected to dendritic cells (DCs) to make an MHC-independent vaccine against BCSCs in TNBC (181). There are reports that CAR-T-CD133 cell therapy presented favorable efficacy in the treatment of human solid cancers, such as breast cancer, in a clinical trial (NCT02541370) [219,220]. The overexpression of epithelial cellular adhesion molecule (EpCAM) in BCSCs provides a potential target for engineering CAR-T-EpCAM cells [221,222]. CAR-T-EpCAM cell therapy has shown significant anti-tumor activity in prostate cancer in a preclinical study [223]. In a clinical trial, CAR-T-EpCAM cell therapy has been evaluated for its anti-tumor efficiency and safety (NCT02915445).

#### 4.4.3. Inhibition of Immune Signaling Receptors

CXCR1/2 are the receptors of IL-8 and can stimulate BCSC self-renewal. Additionally, HER2 expression is regulated by CXCR1/2. Therefore, single CXCR1/2 inhibition and combination therapy, along with HER2 inhibition, are potential treatment strategies for breast cancer. Reparixin is a CXCR1/2 inhibitor that specifically targets BCSCs, and has been shown to suppress metastasis in a human xenograft and metastatic TNBC in a preclinical study when combined with paclitaxel [216]. There are two clinical trials in which reparixin plus paclitaxel are used to treat HER2-negative breast cancer or TNBC (NCT02001974, NCT02370238). CD47 overexpression is regulated by HIF-1α and helps BCSCs escape phagocytosis by macrophages [171,224]. Anti-CD47 plus trastuzumab therapy can block HER2-positive breast cancer and overcome resistance to single trastuzumab treatment [225]. There are several clinical trials ongoing that present well-tolerated safety and good pharmacokinetics (NCT03990233, NCT04097769, NCT05076591) (Table 5).

## 5. Conclusions and Perspectives

The accumulative studies have shown that BCSCs are responsible for treatment resistance, cancer relapse, and metastasis. At the molecular level, BCSCs are governed by multiple signaling pathways, including the Wnt signaling pathway, Notch signaling pathway, HH signaling pathway, NF-κB signaling pathway, and others. These pathways maintain persistent and aberrant activation in BCSCs to support their self-renewal, differentiation, and dissemination abilities. At the cellular level, BCSCs can interact with different cell populations in the TME, including adipocytes, fibroblasts, endothelial cells, and immune cells. The cell populations and their derived cytokines/chemokines can either contribute to BCSC activity or be influenced by BCSCs to form an inflammatory, hypoxic, and immunosuppressive TME from which BCSCs can benefit.

Although most breast cancer patients in the early stages show a good prognosis, some patients experiences drug resistance and cancer relapse [226,227]. Even worse, breast cancer dissemination is a possibility because quiescent metastatic BCSCs escape treatment [228]. The metastatic BCSCs exhibit great plasticity between epithelial cells and mesenchymal cells, providing them with the unique capability to survive, expand, self-renew, and differentiate, all of which contribute to breast cancer progression. Cancer cells undergoing EMT seem to be a prerequisite for BCSC induction, with increased frequency of CD44^+^CD24^−^ cells. After BCSCs finish disseminating and reseeding to secondary organs, they can revert to transform into an epithelial-like phenotype (MET) for colonization and macrometastasis or remain dormant for decades to evade drug treatment, likely in the mesenchyme-like form [229]. This unique plasticity of BCSCs hides them from treatment and causes a lower response to treatments for metastatic/distant relapse [230]. Furthermore, there is still no clear boundary to distinguish the change in signaling pathways between BCSCs and non-stem-like breast cancer cells [231]. A lot of studies have proved that the main signaling pathways, Wnt signaling, Notch signaling, and HH signaling, are overall activated in breast cancer. This means under a stressful environment, non-stem-like breast cancer cells may transform into BCSCs to promote tumorigenesis [232,233,234,235]. Additionally, based on the theory that BCSCs originate from cancerous normal stem cells, BCSCs also share similar features with normal stem cells, such as quiescence and immune tolerance. Therefore, developing therapeutic strategies to specifically track and target BCSCs is one of the pressing needs in breast cancer treatment.

In terms of therapeutic strategy development, early intervention is more efficient in eliminating bulk cancer cells. The data from the last 10 years has shown that around 10% of women with early-stage breast cancer experience a late recurrence after treatment [236,237]. Hence, the optimization of current therapies is urgently needed. First, strategic combination therapies should be considered in preclinical and clinical research. BCSC numbers and functions are regulated by different mechanisms. Some of the signaling pathways have overlapping functions in BCSCs and may compensate each other. Research is warranted to understand how these signaling pathways work in BCSCs, which will provide a rationale to target various key signaling pathways in BCSCs. For example, only using anti-HER2 therapy, Trastuzumab, for HER2-positive breast cancer patients fails to eradicate all BCSCs and ultimately leads to the development of therapeutic resistance [238,239]. Some studies have shown that Trastuzumab can overcome the resistance and prevent BCSC self-renewal when combined with other drug treatments, such as Imetelstat—a telomerase inhibitor, NVP-BKM120—a PI3K inhibitor, and deBouganin—a protein synthesis inhibitor [240,241,242]. Secondly, developing immunotherapy is a promising direction to specifically target BCSCs, such as the CAR-T cell therapy. For example, γδ T cells upregulate the antigen-presenting ability of BCSCs and thereby increase their sensitivity to CD8 T-cell killing via an MHC-restricted manner in a mouse model [243]. This study inspires the development of γδ T-cell therapy to eliminate BCSCs in the future. In summary, aberrant molecular signaling pathways and cellular interactions support BCSC activities and help BCSCs to survive under an immune surveillance environment. Therefore, developing specific therapeutic strategies to eliminate BCSCs will be a promising direction in breast cancer treatment.

## Figures and Tables

**Figure 1 cancers-14-03287-f001:**
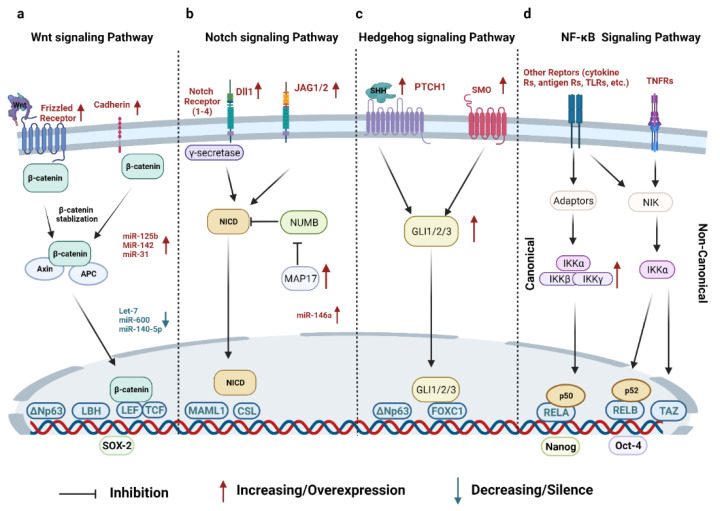
Summary of important signaling pathways in BCSCs. (**a**) The Wnt signaling pathway is activated when Wnt ligands bind to the receptor complex, which dissociates β-catenin from the destruction complex and activates β-catenin as the transcriptional co-activator within the LEF/TCF transcriptional complex. In BCSCs, several frizzled receptors are upregulated and several miRNAs are altered in expression. (**b**) The Notch signaling pathway is induced by juxtacrine interactions between Dll or JAG1/2 and Notch receptors, resulting in the release of NICD as the transcription factor for activation of a panel of genes. The expression of Dll and JAG1/2 ligands can be elevated from several cellular sources within the BCSC microenvironment to induce juxtacrine activation of Notch signaling in BCSCs. MAP17 inhibits NUMB, the antagonist of NICD, to activate Notch signaling pathway in BCSCs. (**c**) The activation of HH signaling pathway initiates from HHs binding to the PTCH receptor, leading to the de-repression of SMO. The GLI 1/2/3 translocate to the nucleus and form the transcriptional complex to activate gene expression. In BCSCs, PTCH1, SMO, and GLIs are upregulated to enhance the HH signaling pathway. (**d**) The canonical NF-κB signaling pathway is initiated from various receptors including TNFR, which activates the trimeric IKK complex via several steps. The IKK complex induces the ubiquitin-dependent degradation of IκBα, which results in the nuclear translocation of canonical NF-κB dimer RelA/p50. The non-canonical NF-κB signaling pathway can also be initiated by several TNFR superfamily members such as RANKL. NIK and IKKα are required for propagating non-canonical signaling pathways. This process induces the degradation of p100, leading to the generation of the RELB/p52 dimer that translocates to the nucleus. In BCSCs, the activation of the NF-κB signaling pathway is associated with BCSC self-renewal and the expression level of BCSC markers. Moreover, some crucial transcription factors, such as Oct-4, Nanog, and SOX2, are consistently activated in BCSCs to maintain their self-renewal capacity. Key abbreviations: APC, adenomatous polyposis coli; LBH, limb bud-heart; TCF/LEF, T-cell factor/lymphoid enhancer factor; NICD, Notch intracellular domain; MAML1, mastermind-like transcriptional co-activator 1; CSL, CBF1, suppressor of hairless, lag-1; GLI 1/2/3, GLI Family Zinc Finger 1/2/3; FOXC1, forkhead box C1; NIK, NF-κB-inducing kinase; IKK, inhibitor of κB kinases.

**Figure 2 cancers-14-03287-f002:**
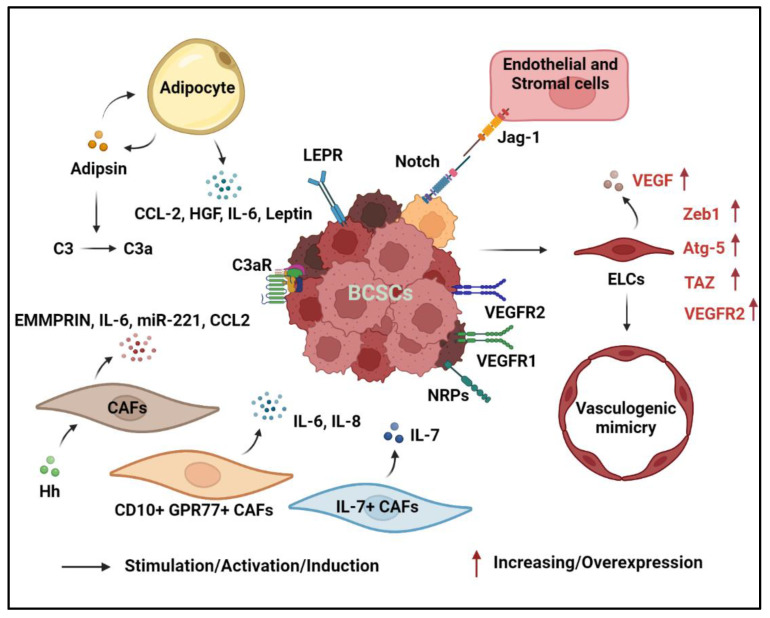
Crosstalk between BCSCs and stromal cells in the TME. Within the TME, various cell types are known to secrete various cytokines or chemokines such as IL-6, IL-8, CCL2, IL-7, VEGF, etc., that can directly or indirectly influence BCSC function. In addition, different cells within the TME also can promote BCSC expansion via lineage-specific factors. For example: adipocytes secrete adipokine adipsin and leptin that are shown to promote BCSCs; Notch ligands can be produced by endothelial cells, normal epithelial or non-CSC cancer cells, or other stromal cells to promote juxtacrine activation of Notch pathway in BCSCs; and BCSCs can differentiate into endothelial-like cells, which initiates blood vessel formation—a process known as vasculogenic mimicry—and ultimately supports tumor growth.

**Figure 3 cancers-14-03287-f003:**
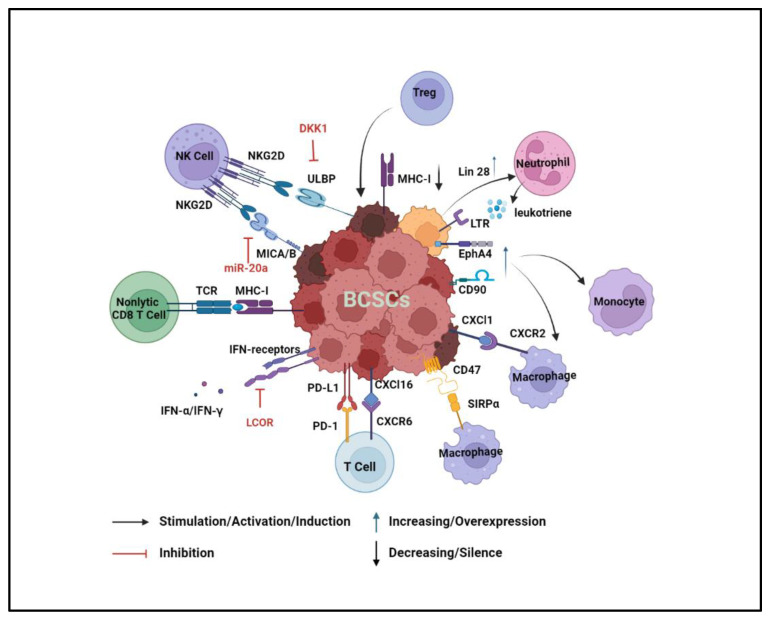
Crosstalk between BCSCs and immune cells in the TME. BCSCs can escape immune surveillance by regulating their surface ligands to suppress the immune response. The expression of ULBP and MICA/B is reduced to suppress the binding with NK cell receptors and evade NK cell-mediated cell killing. Antigen-presenting capacity is decreased to prevent the effector T-cell activation. PD-L1 and CD47 are overexpressed in BCSCs to suppress the anti-tumor effects of T cells and macrophages respectively. Additionally, BCSCs can recruit nonlytic CD8 T cells, Treg cells, and neutrophils to enhance self-renewal and metastatic abilities. Last but not least, an immunosuppressive niche is constituted by recruiting and activating the immunosuppressive cell population, including TAM, Treg cells, and neutrophils.

**Table 1 cancers-14-03287-t001:** BCSC Surface Markers.

**Human**	CD44+ [13]
	Aldehyde dehydrogenase 1 (ADLH1) [22]
	ATP-binding cassette subfamily G member 2 (ABCG2) [27]
	CD133 [28]
	CD49f [29]
	Leucine-rich repeat-containing G protein-coupled receptor 5 (LGR5) [30]
	Stage-specific embryonic antigen 3 (SSEA-3) [31]
	CD70 [32]
	Protein C receptor (PROCR) [33]
	Nectin-4 [34]
	EpCAM [35]
	CD90 [36]
**Mouse**	CD29 [37]
	CD61 [38]

**Table 2 cancers-14-03287-t002:** Targeting BCSCs in Clinical Trials.

BCSCs Marker	Agent/Intervention	Study Phase	Clinicaltrials.gov Identifier	Study Status	Type/Stage of Breast Cancer
CD44	Bivatuzumab Mertansine (CD44v6)	I	NCT02254005	Completed	Breast Neoplasms
CD133	Other: Immunohistochemistry Staining Method	N/A	NCT04873154	Recruiting	Breast Cancer
ALDH1	Other: Immunohistochemistry Staining Method and Laboratory Biomarker Analysis	N/A	NCT00949013	Completed	Early-Stage Breast Cancer
	Other: Immunohistochemistry Staining Method and Laboratory Biomarker Analysis	N/A	NCT01424865	Unknown	Breast Cancer
	Doxorubicin-Cyclophosphamide Regimen	N/A	NCT04581967	Recruiting	Breast Cancer
EGFR/HER2	Trastuzumab	N/A	NCT01424865	Unknown	Breast Cancer

**Table 3 cancers-14-03287-t003:** Targeting the Signaling Pathways of BCSCs in Clinical Trials.

Targeted Signaling Pathway	Agent	Study Phase	Clinicaltrials.gov Identifier	Study Status	Type/Stage of Breast Cancer
Wnt Signaling Pathway	Vantictumab	I	NCT01973309	Completed	Metastatic HER2-Negative Breast Cancer
	Foxy-5	I	NCT02020291	Completed	Metastatic Breast Cancer
	Cirmtuzumab with Paclitaxel I	I	NCT02776917	Active, not Recruiting	Breast Neoplasms
	LGK974	I	NCT01351103	Recruiting	TNBC
Notch Signaling Pathway	MK-0752 (GSI)	I	NCT00106145	Completed	Advanced Breast Cancer
	MK-0752 (GSI) with Docetaxel	I/II	NCT00645333	Completed	Metastatic Breast Cancer
	PF-03084014	II	NCT02299635	Terminated	Advanced-Stage TNBC
	PF-03084014 with Docetaxel 260	I	NCT01876251	Terminated	Advanced-Stage TNBC
	Melatonin with Vitamin D	II	NCT01965522	Completed	Early-Stage Breast Cancer
Hedgehog Signaling pathway	Vismodegib with Neoadjuvant Paclitaxel, Cyclophosphamide, and Epirubicin	II	NCT02694224	Recruiting	TNBC
	Taladegib	I	NCT02784795	Completed	Metastatic Breast Cancer
	Itraconazole	Unknown	NCT00798135	Completed	Metastatic/Non-Metastatic Breast Cancer
	Itraconazole with Capivasertib	I	NCT04712396	Completed	Metastatic TNBC/HR2-Positive Breast Cancer

**Table 4 cancers-14-03287-t004:** Targeting the VEGF in the BCSC Microenvironment in Clinical Trials.

Components	Agent/Intervention	Study Phase	Clinicaltrials.gov Identifier	Study Status	Type/Stage of Breast Cancer
Vascular Endothelial Growth Factor (VEGF)	Bevacizumab	II	NCT00016549	Completed	Breast Cancer
			NCT01190345	Completed	
	Bevacizumab with Herceptin	I/II	NCT00095706	Completed	Breast Cancer

**Table 5 cancers-14-03287-t005:** Immunotherapy in Clinical Trials.

Strategies	Agent	Study Phase	Clinicaltrials.gov Identifier	Study Status	Type/Stage of Breast Cancer
Vaccine	Digoxin	II	NCT01763931	Completed	Breast Cancer
	MVA-brachyury-TRICOM	I	NCT02179515	Completed	HER2-Positive; TNBC; Metastatic Breast Cancer
			NCT04296942	Terminated	
			NCT04134312	Completed	
			NCT03384316	Completed	
	Multiantigen DNA Plasmid-based Vaccine (CD105, Yb-1, SOX2, CDH3 and MDM2)	I	NCT02157051	Recruiting	HER2-Negative Breast Carcinoma; Recurrent Breast Carcinoma; Stage III/IIIA/IIIB/IIIC/IV III Breast Cancer
	DCs Pulsed with the Lysate of Aldefluor-Positive Cells	I/II	NCT02063893	Completed	Breast Neoplasms
Chimeric Antigen Receptor (CAR) T cells	Anti-CD133-CAR Vector-Transduced T cells	I/II	NCT02541370	Completed	Breast Cancer
	Anti-EPCAM	I	NCT02915445	Recruiting	Recurrent Breast Cancer
Chemokine Inhibitors	Reparixin and Paclitaxel	I	NCT02001974	Completed	HER2-Negative Metastatic Breast Cancer
	Reparixin and Paclitaxel with Placebo	II	NCT02370238	Completed	Metastatic TNBC
Anti-CD47	BI 765063 and BI 754091	I	NCT03990233	Recruiting	Solid Breast Cancer
	HX009	I	NCT04097769	Active, Not Recruiting	Advanced Malignancies
	IMM2902	I	NCT05076591	Active, Not Recruiting	Advanced Breast Cancer

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
