# Peer review of "Breast Cancer Stem Cells: Signaling Pathways, Cellular Interactions, and Therapeutic Implications"

_cancers, 2022, doi:10.3390/cancers14133287_

Round 1

Reviewer 1 Report

Breast cancer stem cells (BCSCs) are small population of cells exist in the heterogeneity of breast tumor with special characteristics like self-renewal and differentiation. These CSCs plays a major role in breast cancer metastasis, tumor progression and resistance to therapy. In this review, Wang et al. reported that role of various cellular interactions, signaling pathways such as Wnt, Notch, Hedgehog and their pathway specific therapeutic targets for eliminating CSCs. Overall this study summarized very well about breast cancer stem cells with adequate information and depicted in figures and tables. However, a few suggestions can further improve this review.

1. BCSCs will be isolated from heterogenous populations of human and mouse breast cancer cells by using various cell surface markers. In literature, it is known that human (CD44, CD24, CD133, EpCAM and ALDH+) and mouse breast cancer cells (CD29, CD61) express different cell surface markers. It would be more interested to readers if authors include this topic and with a table of CSC specific surface markers to human and mouse. Find the below three papers (titles) for your reference and citation.

(i). Biglycan promotes cancer stem cell properties, NFκB signaling and metastatic potential in breast cancer cells.

(ii) TGFβ1 promotes breast cancer local invasion and liver metastasis by increasing the CD44hi/CD24- subpopulation.

(iii) Inhibiting epidermal growth factor receptor signaling potentiates mesenchymal to epithelial transition of breast cancer stem cells and their responsiveness to anticancer drugs.

2. A review should be included with the latest updates in the field for the interest of scientific community. Transcription factors, Oct-4, Nanog, Sox-2 have been widely studied in the field of CSCs. It is important to include the other pathways and transcription factors which are also important for enrichment and maintenance of CSCs such as EGFR-STAT3, JAK, TGF-β pathways.

3. A minor grammatical changes needs to be corrected.

Author Response

Thanks for the positive review. Here is point-by-point response.

  1. BCSCs will be isolated from heterogenous populations of human and mouse breast cancer cells by using various cell surface markers. In literature, it is known that human (CD44, CD24, CD133, EpCAM and ALDH+) and mouse breast cancer cells (CD29, CD61) express different cell surface markers. It would be more interested to readers if authors include this topic and with a table of CSC specific surface markers to human and mouse. Find the below three papers (titles) for your reference and citation.

    (i). Biglycan promotes cancer stem cell properties, NFκB signaling and metastatic potential in breast cancer cells.

    (ii) TGFβ1 promotes breast cancer local invasion and liver metastasis by increasing the CD44hi/CD24- subpopulation.

    (iii) Inhibiting epidermal growth factor receptor signaling potentiates mesenchymal to epithelial transition of breast cancer stem cells and their responsiveness to anticancer drugs.

Response: We added a paragraph and a table (table 1), from line 73-74 to summarize BCSC markers and added a section to include all the proposed pathways. Added references as suggested as well. Thanks.

  1. A review should be included with the latest updates in the field for the interest of scientific community. Transcription factors, Oct-4, Nanog, Sox-2 have been widely studied in the field of CSCs. It is important to include the other pathways and transcription factors which are also important for enrichment and maintenance of CSCs such as EGFR-STAT3, JAK, TGF-β pathways.

We cited the paper listed here in section 2. We added the new content in 2.1.4 and 2.1.5 (line275-292) to summarize the new signaling pathways.

  1. A minor grammatical changes needs to be corrected.

All authors have read the MS and made some corrections in line 29, 57, 58, 79, 85, 101, 112, 113, 135, 233, 237, 637, 638. Three English speakers in the author list read through and made corrections. Thanks.

Reviewer 2 Report

In this review, the authors have comprehensively reviewed the current knowledge regarding breast cancer stem cell (BCSC) in three sections, from the basic aspect to clinical implications. The content of the review is enriched, and the figures and tables are well-organized. This review could be accepted after the following revisions.

1. The novelty of the review and its innovative opinions from currently published reviews could be emphasized in the Abstract and Introduction;

2. In the second section the intercommunication between BCSCs and stromal cellular components was summarized, while the potential implication could be more precisely stated;

Author Response

Thanks for the positive comments and constructive critiques. Here is the point-by-point response.

1. The novelty of the review and its innovative opinions from currently published reviews could be emphasized in the Abstract and Introduction;
We highlight the novelty and significance of the abstract (line 35-38) and introduction (line 103-105)
2. In the second section the intercommunication between BCSCs and stromal cellular components was summarized, while the potential implication could be more precisely stated;
We highlighted some potential implication in section 3. Lines 338-339, 364-366, 412-415. 

Reviewer 3 Report

The manuscript by Wang et al provides a clear picture of Breast cancer Stem Cells. The authors focus the attention on the signalling pathways activated in these cells and on the crosstalk that they establish with other cell types. Moreover they describe the possible strategies to target these tumorigenic cells. The review is cohomprensive, interesting, well structured and well written. 

Author Response

Thanks for the positive review.

Reviewer 4 Report

A comprehensive review on signaling pathways involved in breast cancer stem cells biology and response to treatments. The review is well-written, complete and with a nice iconography. The part on clinical trial is especially useful.

Author Response

Thanks for the positive review.